# Clinical presentation and outcomes of the first patients with COVID-19 in Argentina: Results of 207079 cases from a national database

Daniel Schönfeld[1], Sergio Arias[2]*, Juan Carlos Bossio[2], Hugo Fernández[2], David Gozal[3], Daniel Pérez-Chada[4]

1 Centro Diagnóstico San Jorge, Puerto Madryn, Chubut, Argentina, 2 Instituto Nacional de Enfermedades Respiratorias "Dr. Emilio Coni", Administración Nacional de Laboratorios e Institutos de Salud "Dr. Carlos G. Malbrán", Santa Fe, Argentina, 3 Department of Child Health, Women and Children's Hospital, University of Missouri, Columbia, MO, United States of America, 4 Servicio de Neumología, Hospital Universitario Austral, Pilar, Buenos Aires, Argentina

* sergioariassfe@gmail.com

**Data Availability Statement:** There are legal restrictions on sharing the dataset. The dataset is

## Abstract

### Background

There is limited evidence on the clinical characteristics of SARS-CoV-2 infection in Latin America. We present findings from a nationwide study in Argentina.

### Research question

What is disease severity measures and risk factors are associated with admission to an intensive care unit and mortality?

### Study design and methods

Data were extracted from the COVID-19 database of the Integrated Argentina Health Information System, encompassing the period of March 3rd to October 2nd, 2020, using a standardized case report form that included information on contact history, clinical signs and symptoms, and clinical diagnosis. Information was collected at the initial site of care and follow-up conducted through calls by the regional healthcare authorities. A confirmed case of COVID-19 was defined as having a positive result through sequencing or real-time reverse-transcriptase polymerase chain reaction (RT-PCR) assay of nasal and pharyngeal swab specimens.

### Results

RT-PCR testing was positive in 738,776 cases. Complete datasets were available for analysis in 207,079 cases. Mean age was 42.9±18.8 years, 50.0% were males. Frequent co-existing conditions included hypertension (19.2%), diabetes (9.7%), asthma (6.1%) and obesity (5.2%). Most common symptoms included fever (58.5%), cough (58.0%), headache (45.4%), and sore throat (42.1%). Death or ICU admission were independently associated with older age, male, coma, dyspnea or tachypnea, and seizures, with underlying co-

an official database of COVID 19 pandemic in Argentina and authorization about use, analysis and public diffusion of this date, can only be approved by the Argentinian government. The contact information of a data access is the "Dirección Nacional de Epidemiología e Información Estratégica" – email: nuevosnvs2@gmail.com. Information Access might be requested from the Instituto Nacional de Enfermedades Respiratorias Emilio Coni – email: direccionconi@gmail.com.

**Funding:** "Centro Diagnóstico San Jorge. Puerto Madryn" did not play a role in the study design, data collection and analysis, decision to publish, or preparation of the manuscript and only provided financial support in the form of one of authors' salaries. The author contributions roles were reviewed and accurately indicated. The funder provided support in the form of salaries for authors: a. Centro Diagnóstico San Jorge pay salary to Daniel Schönfeld. b. Instituto Nacional de Enfermedades Respiratorias "Dr. Emilio Coni", Administración Nacional de Laboratorios e Institutos de Salud "Dr. Carlos G. Malbrán" pay salaries to Sergio Arias, Juan Carlos Bossio and Hugo Fernández. c. Hospital Universitario Austral pay salaries to Daniel Pérez-Chada. d. Department of Child Health, Women and Children's Hospital, University of Missouri, pay salary to David Gozal. The funders did not have any additional role in the study design, data collection and analysis, decision to publish, or preparation of the manuscript. The specific roles of these authors are articulated in the 'author contributions' section.

**Competing interests:** Commercial affiliation: This does not alter our adherence to PLOS ONE policies on sharing data and materials.

morbidities such as immunodeficiency, chronic renal failure, and liver disease showing the strongest effects.

## Interpretation

Most cases of COVID-19 diagnosed in Argentina were mild and had a favorable outcome, but fatality rates were relatively elevated. Risk factors for adverse outcome included older age, male sex, coma and seizures, and the concurrent presence of several morbidities. These data may be useful for healthcare providers and healthcare policy makers of low-middle income and Latin American countries to guide decisions toward optimized care during the pandemic.

## Introduction

On December 31, 2019, an outbreak of pneumonia caused by a novel coronavirus (SARS-CoV-2) was reported in the city of Wuhan, China [1]. Since then, coronavirus 19 disease (COVID-19) spread rapidly across different continents with a large number of people being infected in a short period of time, thereby challenging the healthcare system capacity and resources throughout the planet, such that on March 11th, 2020, the World Health Organization (WHO) declared COVID-19 as a pandemic [2].

Countries adopted different response strategies according to the speed of contagion and local characteristics of affected cases [3]. Intriguingly, significant differences have been reported in the demographics, clinical features, admission rates, need for intensive care admission, and overall outcomes among different countries and settings [4–8]. The arrival of the virus to Latin America posed a great challenge and became a focus of attention and concern for the WHO [9]. The first case of COVID-19 in Argentina was reported on March 3rd, 2020. A national lockdown was imposed on March 20th, 2020 with various levels of implementation across the country, and is still in effect at the time of submission. To contain the COVID-19 spread, the government implemented a national lockdown as of March 20th, 2020, with various levels of implementation across the country, and is still ongoing at the time of submission. On July 31st, 2020, the Ministry of Health released a report stating the reinforcement of the health system by increasing the number of ICU beds by 40%, including professionally trained staff and critical care support infrastructure. Twelve new modular hospitals were opened in the geographic areas where most COVID-19 cases seemed to be concentrated. On January 10th in 2021, the Ministry of Health reports indicated a total of 1,714,409 confirmed cases, with 1,504,330 patients having recovered and 44,417 had died.

Most of the available evidence on COVID-19 characteristics and clinical features has been based on studies of hospitalized patients, with only a few reports based on population wide datasets [10–13]. Furthermore, there is only a paucity of studies reporting the clinical characteristics and outcomes of patients in Latin America [11, 14–16]. The COVID-19 pandemic represents a significant challenge particularly in low to middle-income countries (LMIC) of the region, as interventions such as social distancing are challenging if not virtually impossible to implement in large overcrowded urban areas, and issues such as concurrent dengue outbreaks and limited testing capacity may further impede efforts to stop the spread [17]. Understanding features associated with COVID-19 susceptibility and adverse prognostic factors is therefore crucial to guide local health authorities in their quest to allocate their resources more efficiently and avoid over-stressing the already constrained healthcare system.

The goal of the present exploratory study is to describe the clinical characteristics and severity of disease at the time of their initial evaluation of a large cohort of patients diagnosed with COVID-19 over the initial 6 months since the first case was declared in Argentina, and to report on patient outcomes while assessing for potential underlying risk factors associated with admission to an intensive care unit (ICU) or with death.

## Methods

### Data sources

The COVID-19 database of the Integrated Argentine Health Information System (Sistema Integrado de Información Sanitaria Argentina (SIISA)) which includes all recorded cases in Argentina from March 3rd to October 2nd, 2020. SIISA uses a case report form based on the revised tool provide by the WHO for confirmed Novel Coronavirus COVID-19. In addition, the SIISA form includes information on contact history, clinical signs and symptoms, and clinical diagnosis as defined by the attending physicians at the time of initial evaluation (clinical and radiological evidence of pneumonia, severe pneumonia and presence of respiratory failure) [18]. The case report form used herein is provided in the S1 Appendix. Data were collected at the initial site of care and regional healthcare authorities conducted follow up assessments via telephone calls.

### Study definitions

Only cases meeting a definition of suspected COVID-19 (as of April 16th, 2020 the current case definition includes two or more of the following symptoms: fever $\geq 37.5°C$, cough, odynophagia, shortness of breath, anosmia or dysgeusia) were subjected to testing as per guidelines issued by the National Ministry of Health in Argentina. A confirmed case of COVID-19 was defined as having a positive result through either sequencing or real-time reverse-transcriptase–polymerase-chain-reaction (RT-PCR) assay of nasal and pharyngeal swab specimens. Only laboratory-confirmed cases were included in the analysis. The presence of comorbidities was evaluated by the attending physicians using no standardized definitions. The primary composite endpoints were admission to an intensive care unit (ICU) or death.

### Study oversight

The study database was obtained and processed by the National Institute of Respiratory Diseases "Dr. Emilio Coni" which acted as data custodian. The study protocol was approved by the Independent Review Board of Hospital Zonal de Trelew, Chubut. According to National Law 25326. In the framework of the COVID-19 pandemic, written consent was waived for this type of epidemiological studies16, whereby data were anonymized to preserve confidentiality, analyzed and interpreted by the authors. All the authors reviewed the manuscript and checked for the accuracy and completeness of the data as well as for the adherence of the study to the protocol.

### Statistical analysis

Continuous variables were described as means and standard deviation (SD), medians and interquartile ranges (IQR). Categorical variables were presented as absolute values and percentages. Continuous variables were compared using the independent group t tests (normal distribution). Categorical variables were compared by $\chi$-square tests. The association between demographic characteristics, baseline symptoms, clinical diagnosis and comorbidities with adverse outcomes (admission to ICU or death) were evaluated using Odds Ratios (OR).

Initially, an unadjusted (univariate) analysis was performed between variables and the statistical significance of the OR was evaluated using $\chi$-square tests. Subsequently, associations were analyzed adjusting all the variables by binary logistic regression. Covariates significantly associated with increased risk in the univariate analysis were included in a multivariate logistic regression analysis. A p-value <0.05 was considered statistically significant. Data analysis was performed using SPSS ® version 25.0 software (SPSS Inc., Chicago, IL, United States).

## Results

### Epidemiological characteristics

During the study period, 2,078,326 subjects fulfilled the definition of a suspected case. Of these, 1,919,918 (92.4%) were tested by rt-PCR, and a positive result was recorded in 738,776 cases (i.e., positivity rate: 38.5%). Complete datasets were available for 207,079 COVID-19 cases (C1: 28.1%) (Fig 1). When comparing the data from the total number of patients with a positive SARS-CoV-2 test, but with missing data in key variables (C2) with C1 cases, the latter were older (mean age: 42.9 vs. 38.6 years, p < 0.001), were less frequently male (50.0% vs. 51.3%, p < 0.001) and were more likely to be severe cases (hospital admission: 20.1% vs. 5.7%, p < 0.001, ICU admission: 2.7% vs. 0.5%, p < 0.001, deaths: 5.3% vs. 1.8% p < 0.001) (Table 1).

Baseline characteristics of C1 cases are presented in Table 2. The vast majority of cases (84.2%) which tested positive (n = 174,300) lived in the city of Buenos Aires and in the province of Buenos Aires. The initial evaluation was performed in a public healthcare facility in 65.8% of the cases (n = 136,313). Infections were deemed attributable to community transmission in 103,059 (57.9%) and to close contacts in 51,573 (29.0%). A history of travel abroad was present in 0.37% (n = 653). Healthcare workers accounted for 10.6% of C1 cases (n = 18,809).

### Demographics and comorbidities

The mean age of the cohort was 42.9±18.8 years, with 80.7% being younger than 60 years of age. The sample comprised 50.0% men and 50.0% women. At least one underlying disease was reported in 41.0% (n = 84,916). Hypertension was the most frequent coexisting disorder (19.2%), followed by diabetes (9.7%), asthma (6.1%) and obesity (5.2%). Current tobacco smoking was reported by 2.0%, and 2.6% were former smokers (Table 2).

### Symptoms at initial presentation

The most common symptoms were fever (58.5%), cough (58.0%), headache (45.4%), and sore throat (42.1%). General symptoms such as fatigue (32.9%) and myalgia (27.0%), were less common. Anosmia was reported in 25.7% and dysgeusia in 18.9%, and these symptoms were included in the definition of suspected cases as of June 8th, 2020. Gastrointestinal symptoms occurred in 21% (diarrhea 9.9%, abdominal pain 4.5%, vomiting 3.9%, anorexia 2.7%). Neurological complaints including confusion (1.1%), irritability (1.1%), seizures (0.2%) and coma (0.2%) were rare (Table 2).

### Time course of disease evolution

Median time elapsed between initial symptoms and database entry was 4.2 days (IQR: 2–5 days). Median time to ICU admission was 5.8 days (IQR: 1–8 days) and median time to death was 16 days (IQR: 7–21 days). Median time of days between ICU admission and death was 11.7 days (IQR: 4–16 days).

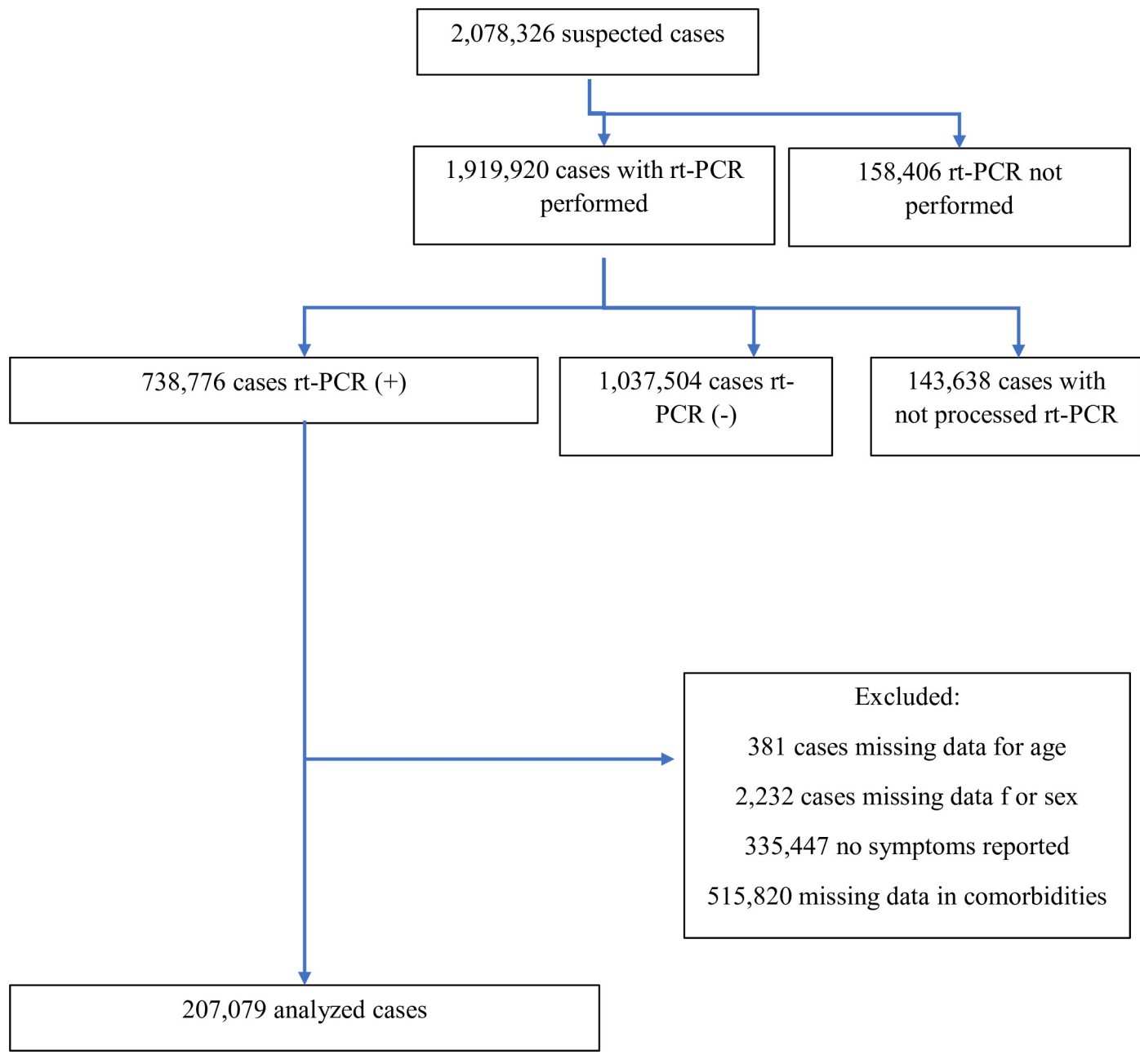

**Fig 1. Flowchart of inclusion of cases.**

### Disease outcomes and associated factors

Admission to ICU occurred in 2.7% of C1 cases (n = 5,652) while the total case fatality rate was 5.3% (n = 10,913). Those cases admitted to general wards and ICU were older (54.2±22.3 years and 63.6±17.4 years, respectively). Mean age of deceased patients was 72.0±14.4 years. Men were more likely to require ICU care (61.9% vs. 38.1%; p<0.05) and to die from COVID-19 (57.4 vs. 42.6%; p<0.05). No comorbidities were reported in 13.9% of cases admitted to ICU and in 9.5% of those who died.

**Table 1. Comparison between included and excluded cases.**

|  | Included–C1 (n = 207,079) | | Excluded–C2 (n = 531,697) | | All Cases (n = 738,776) | | p value (included/excluded) |
|---|---|---|---|---|---|---|---|
|  | No. or Mean | % or SD | No. or Mean | % or SD | No. or Mean | % or SD |  |
| Age | 42.94 | 18.81 | 38.64 | 18.73 | 40.37 | 18.04 | < 0.001 |
| Gender Male | 103,487 | 50.0 | 272,611 | 51.3 | 376,098 | 50.9 | < 0.001 |
| Admitted cases | 41,703 | 20.1 | 30,456 | 5.7 | 72,159 | 9.8 | < 0.001 |
| ICU admitted cases | 5,652 | 2.7 | 2,793 | 0.5 | 8,445 | 1.1 | < 0.001 |
| Deaths | 10,913 | 5.3 | 9,339 | 1.8 | 20,252 | 2.7 | < 0.001 |

Only 3.5% of the whole sample had clinical or radiological characteristics consistent with pneumonia, while 3.6% presented respiratory failure, and only 1.6% appeared as severe pneumonia on admission. These figures increased to 22.0%, 28.6% and 17.5% respectively in ICU-admitted patients. The composite outcome (i.e., ICU admission or death) occurred in 17,995 cases (8.7%). In the univariate analysis, the mean age of patients with adverse clinical course evolution was 69.4±16.4 years vs. 41.1±17.5 years (p < 0.001). Furthermore, patients with unfavorable clinical course presented more frequently with hypertension (52.1%), diabetes (26.8%), and obesity (12.2%) (Table 3).

In the multivariate analysis, age (adjusted odds ratio adjOR: 17.54, 95% CI: 16.13–19.23 for the group >80 years of age; p<0.05) and male gender (adjOR: 1.49, 95% CI: 1.43–1.56, p = 0.001) were associated with adverse composite outcome. Several clinical features during the initial evaluation, such as the presence of coma (adjOR 5.62 95% CI 4.08–7.61), seizures (adjOR: 2.51, 95% CI: 1.84–3.42), dyspnea or tachypnea (adjOR: 2.73 95% CI: 2.57–2.90), and the use of accessory muscles at initial evaluation (adjOR: 2.46 95% CI: 2.19–2.75) were independent predictors of an adverse outcome. In addition, most of the comorbidities were associated with the adverse composite outcome, with highest risk emerging among those with immunodeficiency (adjOR: 2.56 95% CI:2.24–2.91; p<0.05), obesity (adjOR 2.01 95%1.87–2.16; p<0.05), chronic renal disease (adjOR 2.31 95% CI: 2.17–2.60; p<0.05), malignancy (adjOR 2.11 95% CI: 1.93–2.30; p<0.05) and liver disease (adjOR: 2.14 95% CI: 1.65–2.61; p<0.05). In addition, asthma (adjOR: 0.86 95% CI: 0.67–0,95; p = 0.004) was also independently associated with adverse outcomes but interestingly suggestive of a protective effect, while neither current (adjOR: 1.08 95% CI: 0.96–1.23; p = 0.960) nor previous smoking (adjOR 1.00 95%CI: 0.92–1.09; p = 0.202) were associated with increased composite outcomes risk (Table 4 and Fig 2).

## Discussion

The results of this nationwide study among COVID-19 patients from Argentina show that although most of the cases diagnosed with COVID-19 in our database had a favorable outcome, the case fatality rate (CFR), was higher than reported in other studies. However, when we included both C1 and C2 cases, the CFR was similar to those reported in other settings [4–8]. Notwithstanding the fact that the majority of the C1 cases occurred in young subjects with no comorbidities, multivariate analysis identified a cluster of risk factors associated with adverse outcomes with many of these being similar to those reported elsewhere [4]. The most frequent symptoms in this large cohort were fever and cough, followed by headache and odynophagia, with anosmia and dysgeusia being less frequent, and predominantly occurring among milder cases (please note that these two symptoms were only included in the case definition several months later). Nevertheless, hospital admission was required in 20% of the cases, while ICU care was reported in a small percentage of patients, albeit accounting for approximately 50% of the mortality.

**Table 2. Clinical features of cases by level of care and outcome.**

| | All cases (n = 207079) | General ward (n = 41703) | Intensive Care Unit (n = 5652) | Dead (n = 10913) | Resolved (n = 141571) |
|---|---|---|---|---|---|
| | N (%) or median (IQR) | N (%) or median (IQR) | N (%) or median (IQR) | N (%) or median (IQR) | N (%) or median (IQR) |
| **Demographic features** | | | | | |
| Age. years | 41 (2–55) | 55 (37–72) | 66 (54–76) | 74 (63–83) | 38 (28–51) |
| Age. groups | | | | | |
| 0–18 years | 13617 (6.6) | 2094 (5.0) | 118 (2.1) | 25 (0.2) | 10257 (7.2) |
| 15–39 years | 84443 (40.8) | 9602 (23.0) | 364 (6.4) | 249 (2.3) | 64602 (45.6) |
| 40–49 years | 38442 (18.6) | 5597 (13.4) | 526 (9.3) | 505 (4.6) | 28117 (19.9) |
| 50–59 years | 30673 (14.8) | 6043 (14.5) | 971 (17.2) | 1151 (10.5) | 20881 (14.7) |
| 60–69 years | 19788 (9.6) | 6393 (15.3) | 1423 (25.2) | 2370 (21.7) | 11226 (7.9) |
| 70–79 years | 11431 (5.5) | 5960 (14.3) | 1258 (22.3) | 2942 (27.0) | 4387 (3.1) |
| ≥ 80 years | 8685 (4.2) | 6014 (14.4) | 992 (17.6) | 3671 (33.6) | 2101 (1.5) |
| Gender (Male) | 103487 (50.0) | 22183 (53.2) | 3499 (61.9) | 6267 (57.4) | 68886 (48.7) |
| **Baseline symptoms** | | | | | |
| Fever | 121079 (58,5) | 27040 (64,8) | 3632 (64,3) | 6901 (63,2) | 80358 (56,8) |
| Cough | 120183 (58,0) | 25362 (60,8) | 3271 (57,9) | 6414 (58,8) | 80961 (57,2) |
| Sore Throat | 87085 (42,1) | 12151 (29,1) | 964 (17,1) | 1544 (14,1) | 64209 (45,4) |
| Headache | 93939 (45,4) | 12517 (30,0) | 1168 (20,7) | 1893 (17,3) | 70168 (49,6) |
| Fatigue | 68124 (32,9) | 14653 (35,1) | 1790 (31,7) | 3607 (33,1) | 45050 (31,8) |
| Anosmia | 53273 (25,7) | 5893 (14,1) | 380 (6,7) | 447 (4,1) | 41754 (29,5) |
| Myalgia | 55812 (27,0) | 9572 (23,0) | 1002 (17,7) | 1803 (16,5) | 38869 (27,5) |
| Dysgeusia | 39077 (18,9) | 4508 (10,8) | 320 (5,7) | 417 (3,8) | 30560 (21,6) |
| Arthralgia | 19445 (9,4) | 4451 (10,7) | 470 (8,3) | 917 (8,4) | 12496 (8,8) |
| Dyspnea or tachypnea | 10746 (5,2) | 6293 (15,1) | 1502 (26,6) | 2818 (25,8) | 3547 (2,5) |
| Abdominal pain | 9329 (4,5) | 2519 (6,0) | 326 (5,8) | 656 (6,0) | 5777 (4,1) |
| Diarrhea | 20568 (9,9) | 3849 (9,2) | 528 (9,3) | 894 (8,2) | 14281 (10,1) |
| Anorexia | 5677 (2,7) | 2043 (4,9) | 291 (5,1) | 786 (7,2) | 3043 (2,1) |
| Vomiting | 7995 (3,9) | 1922 (4,6) | 272 (4,8) | 448 (4,1) | 1084 (0,8) |
| Conjunctival injection | 3788 (1,8) | 832 (2,0) | 94 (1,7) | 156 (1,4) | 2534 (1,8) |
| Irritability | 2378 (1,1) | 856 (2,1) | 174 (3,1) | 383 (3,5) | 1284 (0,9) |
| Use of accessory muscles | 2544 (1,2) | 1686 (4,0) | 487 (8,6) | 971 (8,9) | 642 (0,5) |
| Confusion | 2181 (1,1) | 1661 (4,0) | 409 (7,2) | 1053 (9,6) | 349 (0,2) |
| Seizures | 435 (0,2) | 223 (0,5) | 60 (1,1) | 80 (0,7) | 182 (0,1) |
| Coma | 326 (0,2) | 158 (0,4) | 87 (1,5) | 116 (1,1) | 118 (0,1) |
| **Clinical diagnosis** | | | | | |
| Clinical presentation or imaging compatible with pneumonia | 7329 (3.5) | 5269 (12.6) | 1241 (22.0) | 2328 (21.3) | 1788 (1.3) |
| Respiratory failure | 7361 (3.6) | 4596 (11.0) | 1616 (28.6) | 2915 (26.7) | 1980 (1.4) |
| Severe Pneumonia | 3305 (1.6) | 2637 (6.3) | 988 (17.5) | 1603 (14.7) | 371 (0.3) |
| **Comorbidities** | | | | | |
| No comorbidities | 122163 (59.0) | 14584 (35.0) | 783 (13.9) | 1032 (9.5) | 92121 (65.1) |
| Hypertension | 39833 (19.2) | 14128 (33.9) | 2763 (48.9) | 5916 (54.2) | 21763 (15.4) |
| Diabetes | 20058 (9.7) | 7118 (17.1) | 1581 (28.0) | 2961 (27.1) | 10802 (7.6) |
| Obesity | 10854 (5.2) | 3801 (9.1) | 999 (17.7) | 1215 (11.1) | 6100 (4.3) |
| Asthma | 12580 (6.1) | 2359 (5.7) | 269 (4.8) | 396 (3.6) | 8958 (6.3) |
| Previous neurological disease | 5356 (2.6) | 3369 (8.1) | 500 (8.8) | 1680 (15.4) | 1716 (1.2) |
| Heart failure | 5753 (2.8) | 3185 (7.6) | 797 (14.1) | 1729 (15.8) | 2094 (1.5) |

*(Continued)*

**Table 2.** (Continued)

| | All cases (n = 207079) | General ward (n = 41703) | Intensive Care Unit (n = 5652) | Dead (n = 10913) | Resolved (n = 141571) |
|---|---|---|---|---|---|
| | N (%) or median (IQR) | N (%) or median (IQR) | N (%) or median (IQR) | N (%) or median (IQR) | N (%) or median (IQR) |
| Malignancy | 4436 (2.1) | 2148 (5.2) | 403 (7.1) | 974 (8.9) | 1870 (1.3) |
| Chronic Obstructive Pulmonary Disease | 4405 (2.1) | 2296 (5.5) | 548 (9.7) | 1078 (9.9) | 1759 (1.2) |
| Immunodeficiency | 2872 (1.4) | 1120 (2.7) | 230 (4.1) | 337 (3.1) | 1473 (1.0) |
| Pregnancy | 2603 (1.3) | 682 (1.6) | 43 (0.8) | 18 (0.2) | 1730 (1.2) |
| Chronic renal disease | 2340 (1.1) | 1357 (3.3) | 349 (6.2) | 727 (6.7) | 736 (0.5) |
| Liver disease | 914 (0.4) | 397 (1.0) | 84 (1.5) | 185 (1.7) | 409 (0.3) |
| Previous Community-acquired pneumonia | 3156 (1.5) | 1316 (3.2) | 255 (4.5) | 569 (5.2) | 1569 (1.1) |
| Current smoker | 4074 (2.0) | 1133 (2.7) | 241 (4.3) | 389 (3.6) | 2534 (1.8) |
| Former smoker | 5456 (2.6) | 2386 (5.7) | 513 (9.1) | 971 (8.9) | 2595 (1.8) |

Although only a minority of cases had a clinical or radiological diagnosis of pneumonia on admission, a diagnosis of severe pneumonia was associated with poorer prognosis. Patients admitted to ICU were older with a male predominance. Age, comorbidities, and several symptoms on initial evaluation—particularly coma, dyspnea and confusion—were also associated with ICU admission or death.

Our results suggest that ICU admission rate was lower than the pooled 32% summarized in a meta-analysis of 50 studies [19]. However, the ICU mortality rate in this study was high (56.2%). ICU mortality in COVID-19 patients varies widely among the published case series, ranging from 16% to 78% [20]. The CFR of 5.3% reflects the inclusion of cases with complete data, which were also the most severe. A less stringent CFR estimate including all the cases (i.e., C1 and C2) would have resulted in a CFR of 2.7%. Reported CFRs varied from 0.2% in Germany to 7.7% in Italy [21]. The CFR in this study may reflect a younger population with less burden of comorbidities, hospital admissions with a broader case definition, lower pressure on the healthcare systems as cases mostly concentrated in large metropolitan districts with more resilient healthcare systems and expanded hospital bed capacity, attenuated community transmission and improved adherence to lockdown measures during the period covered by this study. However, the ongoing propagation of the outbreak and the ensuing strain on healthcare may lead to radical changes in CFR, since more vulnerable healthcare systems in smaller towns and provinces with less resources may be more adversely impacted and more stressed [22], similar to the observed trends in other countries in the region [23].

The burden of SARS-CoV-2 infection among healthcare personnel has been a matter of concern. The frequency of cases among such workers was 10.6%, higher than initially described for China, but lower than reported in the U.S., Spain and Brazil [24–27]. The emerging occupational risk of contracting COVID-19 in this group is a strong concern for LMIC, considering the need for isolation of the affected personnel and their teammates, which will undoubtedly place an additional strain on the healthcare systems where human resources are already scarce [28].

The frequency of symptoms in our sample was comparable to other studies [29, 30]. Time from symptom onset to consultation was also similar to findings in the UK [6]. Twenty-one percent of the patients presented digestive symptoms upon admission, the most common being anorexia, vomiting, abdominal pain, and diarrhea. Frequency of these symptoms was higher in our series than the pooled prevalence of 15.0% presented in a systematic review of the literature [31].

**Table 3.** Association between clinical features of cases and adverse outcome (Intensive care unit admission or death) (n = 207079, p values < 0.05 unless specified otherwise).

| | No adverse outcome (n = 193690) | Adverse outcome (n = 13389) | Unadjusted odds ratio (OR) |
|---|---|---|---|
| | N (%) or median (IQR) | N (%) or median (IQR) | OR and 95% CI |
| **Demographic features** | | | |
| Age, years | 39 (28–53) | 71 (60–81) | |
| Age, groups | | | |
| 0–18 years | 13487 (7.0) | 130 (1.0) | 0.44 (0.53–0.37) |
| 15–39 years | 83927 (43.3) | 516 (3.9) | 0.28 (0.31–0.25) |
| 40–49 years | 37620 (19.4) | 822 (6.1) | 1 |
| 50–59 years | 28991 (15.0) | 1682 (12.6) | 2.65 (2.89–2.44) |
| 60–69 years | 16885 (8.7) | 2903 (21.7) | 7.87 (8.55–7.25) |
| 70–79 years | 8059 (4.2) | 3372 (25.2) | 19.23 (20.83–17.54) |
| ≥ 80 years | 4721 (2.4) | 3964 (29.6) | 38.46 (41.67–35.71) |
| Gender (Male) | 97982 (50.6) | 7779 (58.1) | 1.42 (1.37–1.47) |
| **Baseline symptoms** | | | |
| Fever | 111571 (57.6) | 8461 (63.2) | 1.26 (1.22–1.31) |
| Cough * | 112344 (58.0) | 7839 (58.5) | 1.02 (0.99–1.06) |
| Sore Throat | 85078 (43.9) | 2005 (15.0) | 0.23 (0.21–0.24) |
| Headache | 91474 (47.2) | 2465 (18.4) | 0.25 (0.24–0.26) |
| Fatigue * | 63731 (32.9) | 4393 (32.8) | 1.00 (0.96–1.03) |
| Anosmia | 52614 (27.2) | 659 (4.9) | 0.14 (0.13–0.15) |
| Myalgia | 53539 (27.6) | 2273 (17.0) | 0.54 (0.51–0.56) |
| Dysgeusia | 38494 (19.9) | 583 (4.4) | 0.18 (0.17–0.20) |
| Arthralgia | 18309 (9.5) | 1136 (8.5) | 0.89 (0.83–0.95) |
| Dyspnea or tachypnea | 7345 (3.8) | 3401 (25.4) | 8.64 (8.26–9.04) |
| Abdominal pain | 5838 (3.0) | 791 (5.9) | 1.36 (1.26–1.47) |
| Diarrhea | 19442 (10.0) | 1126 (8.4) | 0.82 (0.77–0.88) |
| Anorexia | 4783 (2.5) | 894 (6.7) | 2.83 (2.63–3.04) |
| Vomiting | 7406 (3.8) | 589 (4.4) | 1.16 (1.06–1.26) |
| Conjunctival injection | 3584 (1.9) | 204 (1.5) | 0.82 (0.71–0.95) |
| Irritability | 1935 (1.0) | 443 (3.3) | 3.39 (3.05–3.77) |
| Use of accessory muscles | 1414 (0.7) | 1130 (8.4) | 12.53 (11.57–13.58) |
| Confusion | 986 (0.5) | 1195 (8.9) | 19.15 (17.57–20.88) |
| Seizures | 321 (0.2) | 114 (0.9) | 5.17 (4.17–6.41) |
| Coma | 180 (0.1) | 140 (1.0) | 11.85 (9.52–14.75) |
| **Clinical diagnosis** | | | |
| Clinical presentation or images compatible with pneumonia | 4565 (2,4) | 2764 (20,6) | 10,78 (10,24–11,34) |
| **Respiratory failure** | 3873 (2,0) | 3488 (26,1) | 17,27 (16,42–18,15) |
| Severe Pneumonia | 1383 (0,7) | 1922 (14,4) | 23,31 (21,70–25,04) |
| Comorbidities | | | |
| No comorbidities | 120679 (62,3) | 1484 (11,1) | 0,08 (0,07–0,08) |
| Hypertension | 32852 (17,0) | 6981 (52,1) | 5,33 (5,15–5,53) |
| Diabetes | 16470 (8,5) | 3588 (26,8) | 3,94 (3,78–4,11) |
| Obesity | 9227 (4,8) | 1627 (12,2) | 2,77 (2,62–2,92) |
| Asthma | 12053 (6,2) | 527 (3,9) | 0,62 (0,56–0,67) |
| Previous neurological disease | 3501 (1,8) | 1855 (13,9) | 8,74 (8,23–9,27) |
| Heart failure | 3740 (1,9) | 2013 (15,0) | 8,99 (8,49–9,52) |
| Malignancy | 3322 (1,7) | 1114 (8,3) | 5,20 (4,85–5,58) |

(*Continued*)

**Table 3.** (Continued)

| | No adverse outcome (n = 193690) | Adverse outcome (n = 13389) | Unadjusted odds ratio (OR) |
|---|---|---|---|
| | N (%) or median (IQR) | N (%) or median (IQR) | OR and 95% CI |
| Chronic Obstructive Pulmonary Disease | 3121 (1,6) | 1284 (9,6) | 6,48 (6,05–6,93) |
| Immunodeficiency | 2441 (1,3) | 431 (3,2) | 2,61 (2,35–2,89) |
| Pregnancy | 2550 (1,3) | 53 (0,4) | 0,30 (0,23–0,39) |
| Chronic renal disease | 1482 (0,8) | 858 (6,4) | 8,88 (8,15–9,68) |
| Liver disease | 703 (0,4) | 211 (1,6) | 4,40 (3,76–5,13) |
| Previous Community-acquired pneumonia | 2486 (1,3) | 670 (5,0) | 4,05 (3,71–4,42) |
| Current smoker | 3586 (1,9) | 488 (3,6) | 2,01 (1,82–2,21) |
| Former smoker | 4304 (2,2) | 1161 (8,7) | 4,18 (3,91–4,47) |

* p > 0.05

Presence of tachypnea, use of accessory muscles, a clinical picture of pneumonia, or the presence of respiratory failure or severe pneumonia were indicative of poor prognosis. Similarly, neurological symptoms such as mental confusion, coma, and seizures were also associated with higher mortality. The presence of encephalopathy in COVID-19 has been attributed to tissue hypoxia, metabolic derangements with agitation and confusion, and these were reported in 5.1% of cases [32]. In this study the prevalence of confusion, coma and seizures was 1.5%. In our series, only 3.5% of cases had radiologic or clinical evidence of pneumonia on initial evaluation. While a majority of COVID-19 patients presented with mild illness and are not diagnosed with pneumonia, opacities in CT have been reported in up to 55.4% of non-critically ill patients [10, 33]. Previous case series have reported that computed tomography is more sensitive than chest X-ray for the detection of pulmonary involvement [34]. Initial chest X-ray may be normal in up to 63% of patients who may later develop clinical or radiological signs of pneumonia [35]. The low frequency of diagnosis of pneumonia may reflect a limited use of CT scans in the evaluation of mild cases in our clinical settings. However, the diagnosis of pneumonia on initial evaluation was associated with a poorer time course of the disease and was present in 22.0% of the patients admitted to ICU.

An association between older age and adverse clinical outcomes has been uncovered very soon in the course of the pandemic, being reported already in the first published studies. In the present study, the risk associated with older age (particularly in those > 80 years) was prominently higher than previously published and needs to be compared with results from other LMIC [6, 12, 13].

We should also point out that even though most of COVID-19 cases reported no comorbidities, some patients with no risk factors required ICU admission or died. The factors associated with such unique susceptibility are unclear and may reflect genetic elements altering the immune response to the virus [36, 37]. Hypertension, obesity, diabetes, and smoking were present in a lower proportion than findings from local population-based studies. The National Risk Factor Survey found a self-reported prevalence of hypertension of 34.7%, as well as 25.3% for obesity, 12.7% for diabetes and 22.5% for current smokers [38]. It is plausible that the exposed population during a period of national lockdown was composed mainly of those employed in essential activities, and this group would therefore be younger and suffer from less comorbidities than the general population. However, the possibility of underreporting of previous underlying conditions cannot be excluded with complete certainty.

As previously described, patients with an unfavorable clinical evolution were more likely to have a history of immunosuppression, obesity, hypertension, diabetes, cardiac, asthma, hepatic or renal failure, neurological diseases, cancer and COPD [6]. After an adjusted analysis, the

**Table 4. Multivariable logistic regression modelling for the association between clinical features of cases and adverse outcome (Intensive care unit admission or death) (n = 207079).**

| | Adjusted odds ratio (OR and 95% CI) | P value |
|---|---|---|
| Age groups | | |
| 0–18 years | 0.47 (0.57–0.39) | < 0.001 |
| 15–39 years | 0.35 (0.40–0.32) | < 0.001 |
| 40–49 years | Reference | |
| 50–59 years | 2.07 (1.89–2.26) | < 0.001 |
| 60–69 years | 4.69 (4.31–5.10) | < 0.001 |
| 70–79 years | 9.17 (8.33–10.00) | < 0.001 |
| ≥ 80 years | 17.54 (16.13–19.23) | < 0.001 |
| Gender (Male) | 1.49 (1.43–1.56) | < 0.001 |
| Fever | 1.47 (1.54–1.41) | < 0.001 |
| Cough | 0.95 (0.90–0.99) | < 0.001 |
| Fatigue | 0.73 (0.69–0.77) | < 0.001 |
| Dyspnea or tachypnea | 2.73 (2.90–2.57) | < 0.001 |
| Abdominal pain | 1.15 (1.27–1.05) | < 0.001 |
| Diarrhea | 0.83 (0.77–0.90) | < 0.001 |
| Anorexia (*) | 1.06 (1.18–0.96) | 0.23 |
| Vomiting | 1.19 (1.33–1.07) | < 0.001 |
| Use of accessory muscles | 2.46 (2.75–2.19) | < 0.001 |
| Confusion | 0.49 (0.45–0.54) | < 0.001 |
| Seizures | 2.51 (3.42–1.84) | < 0.001 |
| Coma | 5.62 (7.81–4.08) | < 0.001 |
| Clinical presentation or images compatible with pneumonia | 2.39 (2.56–2.24) | < 0.001 |
| Respiratory failure | 4.50 (4.81–4.22) | < 0.001 |
| Hypertension | 1.17 (1.23–1.12) | < 0.001 |
| Diabetes | 1.64 (1.72–1.55) | < 0.001 |
| Obesity | 2.01 (2.16–1.87) | < 0.001 |
| Asthma | 0.86 (0.77–0.95) | < 0.001 |
| Previous neurological disease | 1.97 (2.13–1.82) | < 0.001 |
| Heart failure | 1.36 (1.46–1.26) | < 0.001 |
| Malignancy | 2.11 (2.30–1.93) | < 0.001 |
| Chronic Obstructive Pulmonary Disease | 1.23 (1.35–1.13) | < 0.001 |
| Immunodeficiency | 2.56 (2.91–2.24) | < 0.001 |
| Chronic renal disease | 2.31 (2.60–2.07) | < 0.001 |
| Liver disease | 2.14 (2.61–1.75) | < 0.001 |
| Previous community-acquired pneumonia | 1.26 (1.42–1.12) | < 0.001 |
| Former smoker (*) | 1.00 (1.09–0.92) | 0.96 |
| Current smoker (*) | 1.08 (1.23–0.96) | 0.20 |

(*) p values > 0.05

association with these risk factors remained significant in line with the findings of other studies [13, 39, 40]. Our findings suggest that comorbidities and age were independently associated with adverse outcomes, thereby corroborating other studies [11]. Asthma has been found to be prevalent among patients admitted to the hospital for COVID-19 but has not been consistently associated with a poorer prognosis [6, 40–42]. Our results are in line with these findings, albeit pointing to reduced risk for asthmatic patients, as heterogeneously reported for other countries of Latin America as well [43].

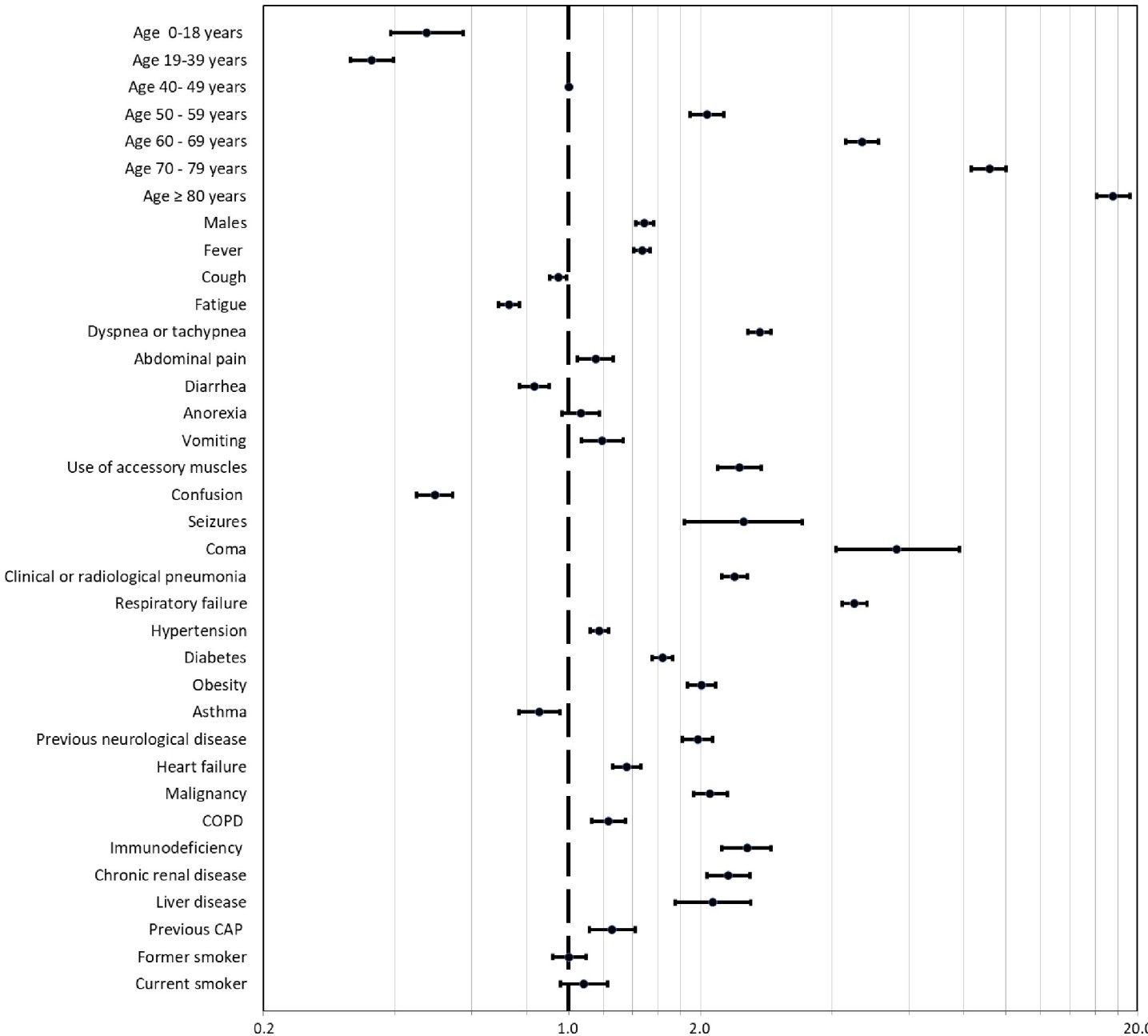

**Fig 2. Factor associated to Covid 19 diseases.** OR and 95% CI. Argentina, March to October 2020.

A history of smoking was not associated with the composite adverse outcome. Meta-analyses have found that current smokers were at greater risk of severe complications, disease progression and a higher mortality rate [44, 45]. Considering that the prevalence of smoking in our population is lower than reported by national surveys (22.5% in adults), smoking status may have been unreported in proportion of our sample yielding underestimates of its potential contribution.

To our knowledge this is one of the first studies reporting the clinical presentation and outcomes of a nationwide sample of COVID-19 patients in South America. It includes a large

number of patients with COVID-19 evaluated at admission and comprises a wide spectrum of severity of presentation, from those with minor symptoms to those requiring admission to the intensive care unit. It presents results from a nationwide population, of diverse social backgrounds, receiving their care in healthcare centers with different access to resources, both public and private, with caregivers with various levels of medical training, scattered throughout the country. It depicts a real-life picture of the clinical presentation and evolution of a disease which is still evolving and fraught with substantial uncertainties.

Our study has some limitations that deserve mention. Information was obtained from the Official Registry System for the COVID-19 pandemic in Argentina. The counts for coronavirus disease deaths are based on mortality data provided by the 24 provinces in Argentina. This process can take several weeks for death records to be confirmed. Therefore, the data shown on this study may be incomplete, especially for the deaths that occurred in the more recent time periods.

Data were collected using a standardized form, and we restricted analyses only to those cases with complete datasets, which represent only 28% of total positive cases. We cannot exclude the possibility that other cases may have not been registered since they may have avoided seeking medical attention. Such possibility could, if of significant magnitude, affect the overall frequency of complications or adverse outcomes related to infection with the virus. In addition, referral and clinical bias could also be a limitation for our study. Indeed, we had no access to hospital record data to include laboratory results or detailed clinical course. Data on symptoms and comorbidities can be incomplete due to the nature of a registry based on point-of-care case report forms. Furthermore, definitions of comorbidities and clinical diagnosis were not standardized. Given that the case definition used to decide whether to perform an rt-PCR test evolved, data on symptoms were subject to variability. Cases with atypical presentations, such as apyrexia or anosmia could have been missed in the initial stages of the pandemic since they were not considered in the case definitions. As previously mentioned, in the early stage of this registry a significant proportion of suspected COVID-19 cases were admitted to hospitals regardless of severity, consequently increasing the proportion of admitted cases. Associations obtained from our regression model should not be considered causal, as some degree of unmeasured confounding and reporting bias can be expected, particularly in the context of a cross-sectional design.

We believe our findings can be generalized for Latin American countries. The predominant route of contagion in Argentina now is community acquired SARS-CoV-2 infection, with IgG seroprevalence values of 53.4% having been reported in urban slum dwellers in the city of Buenos Aires [46].

Our findings should be useful for healthcare providers and healthcare authorities in LMIC and in countries of the Latin America region, as background information for estimates on the evolution of the pandemic. Severe cases can be identified based on the predictors we describe herein, and may help to prioritize attention, make site-of-care decisions, and allocate resources. Risk groups can be protected with tailored measures.

## Supporting information

**S1 Appendix. COVID 19 suspect o confirm case report.** Argentina.
(DOCX)

## Acknowledgments

We are grateful to Dr. Anibal Calmaggi and Dr. Enrique Vázquez Fernández for revision and comments, to Dr. Alejandro Videla for his input in the early version of this study and to the

Library of the School of Biomedical Sciences of Universidad Austral for providing bibliography.

## Author Contributions

**Conceptualization:** Daniel Schönfeld, Sergio Arias, Juan Carlos Bossio, Hugo Fernández, Daniel Pérez-Chada.

**Data curation:** Sergio Arias.

**Formal analysis:** Daniel Schönfeld, Sergio Arias, Hugo Fernández, Daniel Pérez-Chada.

**Investigation:** Daniel Schönfeld, Sergio Arias, Juan Carlos Bossio, Hugo Fernández, Daniel Pérez-Chada.

**Methodology:** Daniel Schönfeld, Sergio Arias, Juan Carlos Bossio, Daniel Pérez-Chada.

**Supervision:** Daniel Schönfeld, Sergio Arias, Juan Carlos Bossio, Hugo Fernández, David Gozal, Daniel Pérez-Chada.

**Validation:** Daniel Schönfeld, Sergio Arias, Juan Carlos Bossio, Hugo Fernández, David Gozal, Daniel Pérez-Chada.

**Visualization:** Daniel Schönfeld, Sergio Arias, Juan Carlos Bossio, Hugo Fernández, David Gozal, Daniel Pérez-Chada.

**Writing – original draft:** Daniel Schönfeld, Sergio Arias, Daniel Pérez-Chada.

**Writing – review & editing:** Daniel Schönfeld, Sergio Arias, David Gozal, Daniel Pérez-Chada.

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
