## [Decision Letter · Decision Letter 0]

4 Jan 2021

PONE-D-20-38991

Clinical presentation and outcomes of the first patients with COVID-19 in Argentina: results of 207079 cases from a national database

PLOS ONE

Dear Dr. Arias,

Thank you for submitting your manuscript to PLOS ONE. After careful consideration, we feel that it has merit but does not fully meet PLOS ONE’s publication criteria as it currently stands. Therefore, we invite you to submit a revised version of the manuscript that addresses the points raised during the review process.

We look forward to receiving your revised manuscript.

Kind regards,

Francesco Di Gennaro

Academic Editor

PLOS ONE

Additional Editor Comments:

Dear authors congratulations for your paper. Follow reviewer suggest to improve your article

Journal Requirements:

2.) Thank you for stating the following in the Financial Disclosure section:

'The author(s) received no specific funding for this work.'

We note that one or more of the authors are employed by a commercial company: Centro Diagnóstico San Jorge. Puerto Madryn

a.) Please provide an amended Funding Statement declaring this commercial affiliation, as well as a statement regarding the Role of Funders in your study. If the funding organization did not play a role in the study design, data collection and analysis, decision to publish, or preparation of the manuscript and only provided financial support in the form of authors' salaries and/or research materials, please review your statements relating to the author contributions, and ensure you have specifically and accurately indicated the role(s) that these authors had in your study. You can update author roles in the Author Contributions section of the online submission form.

b.) Please also provide an updated Competing Interests Statement declaring this commercial affiliation along with any other relevant declarations relating to employment, consultancy, patents, products in development, or marketed products, etc.  

3.) We note that you have indicated that data from this study are available upon request. PLOS only allows data to be available upon request if there are legal or ethical restrictions on sharing data publicly. For information on unacceptable data access restrictions, please see http://journals.plos.org/plosone/s/data-availability#loc-unacceptable-data-access-restrictions.

4.) Please include captions for your Supporting Information files at the end of your manuscript, and update any in-text citations to match accordingly. Please see our Supporting Information guidelines for more information: http://journals.plos.org/plosone/s/supporting-information.

Reviewers' comments:

Reviewer's Responses to Questions

**Comments to the Author**

1. Is the manuscript technically sound, and do the data support the conclusions?

Reviewer #1: Yes

Reviewer #2: Yes

2. Has the statistical analysis been performed appropriately and rigorously? 

Reviewer #1: Yes

Reviewer #2: Yes

3. Have the authors made all data underlying the findings in their manuscript fully available?

Reviewer #1: Yes

Reviewer #2: Yes

4. Is the manuscript presented in an intelligible fashion and written in standard English?

Reviewer #1: Yes

Reviewer #2: Yes

5. Review Comments to the Author

Reviewer #1: Authors wrote a very strong article. The number is incredible. Well done. Only some minor suggestions

1. Introduction: add data on covid burden at the day of resubmission

2. Methods and results no comment.

3. table and figure are clear

4. Disucssion. Compare your data with other paper with high number of patients (see and cite COvid-19 RISk and Treatments (CORIST) collaboration. Common cardiovascular risk factors and in-hospital mortality in 3,894 patients with COVID-19: survival analysis and machine learning-based findings from the multicentre Italian CORIST Study. Nutr Metab Cardiovasc Dis. 2020 Oct 30;30(11):1899-1913. doi: 10.1016/j.numecd.2020.07.031. Epub 2020 Jul 31. PMID: 32912793.

Copy ; doi: 10.1016/j.ejim.2020.08.019. Epub 2020 Aug 25. PMID: 32859477; PMCID: PMC7446618. and doi: 10.1016/j.vph.2020.106805. Epub 2020 Sep 28. PMID: 32992048; PMCID: PMC7521934.)

Add also information on publich health approuch in Argentina to contaoin covid spread.

Congratulations for high quality article

Reviewer #2: This manuscript has described the clinical symptoms of COVID-19 patients at baseline and the clinical characteristics of patients admitted to the ICU. I think the author has used sufficient data and appropriate statistical analysis in making conclusions. The writing is also in standard English and easy to understand. What is interesting and in my opinion is also important in epidemiology, the author also included data on the number of health workers affected by covid19 (paragraphs 242-244) but unfortunately there is no further data on the type of occupation in the patient as a whole. Smoking is also an important data added value in this manuscript because as we all know the number of smokers in developing countries is very high.

6. PLOS authors have the option to publish the peer review history of their article (what does this mean?). If published, this will include your full peer review and any attached files.

Reviewer #1: No

Reviewer #2: No

---

## [Author Response · Author response to Decision Letter 0]

18 Jan 2021

Academics Editor

All manuscript was controlled to adequate to PLOS ONE style requeriments.

2.) Thank you for stating the following in the Financial Disclosure section: 'The author(s) received no specific funding for this work.'

Included

a.) Please provide an amended Funding Statement declaring this commercial affiliation, as well as a statement regarding the Role of Funders in your study.

b.) Please also provide an updated Competing Interests Statement declaring this commercial affiliation along with any other relevant declarations relating to employment, consultancy, patents, products in development, or marketed products, etc.

“Centro Diagnóstico San Jorge. Puerto Madryn” did not play a role in the study design, data collection and analysis, decision to publish, or preparation of the manuscript and only provided financial support in the form of one of authors' salaries.

The author contributions roles were reviewed and accurately indicated.

The funder provided support in the form of salaries for authors [DS, SA, JCB, HF, DG, DPCh], but did not have any additional role in the study design, data collection and analysis, decision to publish, or preparation of the manuscript. The specific roles of these authors are articulated in the ‘author contributions’ section.

Regarding commercial affiliation “This does not alter our adherence to PLOS ONE policies on sharing data and materials”

3.) We note that you have indicated that data from this study are available upon request. PLOS only allows data to be available upon request if there are legal or ethical restrictions on sharing data publicly.

a.) If there are ethical or legal restrictions on sharing a de-identified data set, please explain them in detail (e.g., data contain potentially identifying or sensitive patient information) and who has imposed them (e.g., an ethics committee). Please also provide contact information for a data access committee, ethics committee, or other institutional body to which data requests may be sent.

There are legal restrictions on sharing the dataset. The dataset is an official database of COVID 19 pandemic in Argentina and authorization about use, analysis and public diffusion of this date, can only be approved by the Argentinian government.

The contact information of a data access is the “Dirección Nacional de Epidemiología e Información Estratégica” – email: nuevosnvs2@gmail.com. Information Access might be requested from the Instituto Nacional de Enfermedades Respiratorias Emilio Coni – email: direccionconi@gmail.com.

4.) Please include captions for your Supporting Information files at the end of your manuscript, and update any in-text citations to match accordingly.

Included

Reviewer 1

Reviewer #1: Authors wrote a very strong article. The number is incredible. Well done. Only some minor suggestions

1. Introduction: add data on covid burden at the day of resubmission

Included in reviewed manuscript:

“On January 10th in 2021, the Ministry of Health reports: 1.714.409 confirmed cases, with 1.504.330 patients had recovered and 44.417 died.

2. Methods and results no comment. 

No comment

3. table and figure are clear.

No comment

4. Disucssion. Compare your data with other paper with high number of patients (see and cite COvid-19 RISk and Treatments (CORIST) collaboration. Common cardiovascular risk factors and in-hospital mortality in 3,894 patients with COVID-19: survival analysis and machine learning-based findings from the multicentre Italian CORIST Study. Nutr Metab Cardiovasc Dis. 2020 Oct 30;30(11):1899-1913. doi: 10.1016/j.numecd.2020.07.031. Epub 2020 Jul 31. PMID: 32912793.

Copy ; doi: 10.1016/j.ejim.2020.08.019. Epub 2020 Aug 25. PMID: 32859477; PMCID: PMC7446618. and doi: 10.1016/j.vph.2020.106805. Epub 2020 Sep 28. PMID: 32992048; PMCID: PMC7521934.)

Thanks for the suggestion. Our results about sign and symptoms and comorbidities were compared with many other studies, and their conclusion were included in Discussion section.

However, we appreciate recommendation and include CORIST study in comorbidities risk factor in Discussion and include citation in references (reference 40)

Add also information on publich health approuch in Argentina to contaoin covid spread.

Included in reviewed manuscript:

“To contain the COVID-19 spread, the government implemented a national lockdown as of March 20th, 2020, with various levels of implementation across the country, and is still ongoing at the time of submission.

On July 31st, 2020, the Ministry of Health released a report stating the reinforcement of the health system by increasing the number of ICUs beds by 40%, including professionally trained staff and critical care support infrastructure. Twelve new modular hospitals were opened in the geographic areas were most COVID-19 cases seemed to be concentrated.”

Reviewer 2

Reviewer #2: This manuscript has described the clinical symptoms of COVID-19 patients at baseline and the clinical characteristics of patients admitted to the ICU. I think the author has used sufficient data and appropriate statistical analysis in making conclusions. The writing is also in standard English and easy to understand. What is interesting and in my opinion is also important in epidemiology, the author also included data on the number of health workers affected by covid19 (paragraphs 242-244) but unfortunately there is no further data on the type of occupation in the patient as a whole. Smoking is also an important data added value in this manuscript because as we all know the number of smokers in developing countries is very high.

Unfortunately, we are unable to provide further data on patient’s occupation, other than the health workers. We apologize for this limitation.

Smoking was included precisely in light of the insightful reviewer’s comment. In Argentina, the proportion of smokers is high. However, we found no correlation between smoking and severity of COVID-19.

---

## [Decision Letter · Decision Letter 1]

27 Jan 2021

Clinical presentation and outcomes of the first patients with COVID-19 in Argentina: results of 207079 cases from a national database

PONE-D-20-38991R1

Dear Dr. Arias,

We’re pleased to inform you that your manuscript has been judged scientifically suitable for publication and will be formally accepted for publication once it meets all outstanding technical requirements.

Kind regards,

Francesco Di Gennaro

Academic Editor

PLOS ONE

Additional Editor Comments (optional):

dear authors congratulations

Reviewers' comments:

Reviewer's Responses to Questions

**Comments to the Author**

1. If the authors have adequately addressed your comments raised in a previous round of review and you feel that this manuscript is now acceptable for publication, you may indicate that here to bypass the “Comments to the Author” section, enter your conflict of interest statement in the “Confidential to Editor” section, and submit your "Accept" recommendation.

Reviewer #1: All comments have been addressed

Reviewer #2: All comments have been addressed

2. Is the manuscript technically sound, and do the data support the conclusions?

Reviewer #1: Yes

Reviewer #2: Yes

3. Has the statistical analysis been performed appropriately and rigorously? 

Reviewer #1: Yes

Reviewer #2: Yes

4. Have the authors made all data underlying the findings in their manuscript fully available?

Reviewer #1: Yes

Reviewer #2: Yes

5. Is the manuscript presented in an intelligible fashion and written in standard English?

Reviewer #1: Yes

Reviewer #2: Yes

6. Review Comments to the Author

Reviewer #1: Authors worte an important article on 200.000 patients. Authors improved thier already excellent article that now can be accept

Reviewer #2: No additional suggestions. The large number of subjects will be the reinforcing factor for this article and the authors have written and processed the data well. Good work.

7. PLOS authors have the option to publish the peer review history of their article (what does this mean?). If published, this will include your full peer review and any attached files.

Reviewer #1: No

Reviewer #2: No

---

## [Editor Report · Acceptance letter]

1 Feb 2021

PONE-D-20-38991R1 

Clinical presentation and outcomes of the first patients with COVID-19 in Argentina: results of 207079 cases from a national database 

Dear Dr. Arias:

I'm pleased to inform you that your manuscript has been deemed suitable for publication in PLOS ONE. Congratulations! Your manuscript is now with our production department. 

Kind regards, 

on behalf of

Dr. Francesco Di Gennaro 

Academic Editor

PLOS ONE